# NOVELTYBENCH: Evaluating Language Models for Human-like Diversity

**Yiming Zhang**[*]   **Harshita Diddee**   **Susan Holm**   **Hanchen Liu**
**Xinyue Liu**   **Vinay Samuel**   **Barry Wang**   **Daphne Ippolito**
Carnegie Mellon University

## Abstract

Language models have demonstrated remarkable capabilities on standard benchmarks, yet they struggle increasingly from *mode collapse*, the inability to generate diverse and novel outputs. Our work introduces NOVELTYBENCH, a benchmark specifically designed to evaluate the ability of language models to produce multiple distinct and high-quality outputs. NOVELTYBENCH utilizes prompts curated to elicit diverse answers and filtered real-world user queries. Evaluating 20 leading language models, we find that current state-of-the-art systems generate significantly less diversity than human writers. Notably, larger models within a family often exhibit less diversity than their smaller counterparts, challenging the notion that capability on standard benchmarks translates directly to generative utility. While prompting strategies like in-context regeneration can elicit diversity, our findings highlight a fundamental lack of distributional diversity in current models, reducing their utility for users seeking varied responses and suggesting the need for new training and evaluation paradigms that prioritize diversity alongside quality.[1]

## 1 Introduction

Diversity of opinion, preference, and experience is a fundamental trait of being human. If you were to ask "Tell me a joke" or "What is the best book of all time?" to the next five people you talk to, with high likelihood you would receive five different answers. It is reasonable to expect language models to generate responses that have the same level of diversity as humans. Yet, when we ask models such as GPT-4 to recommend a movie or Claude 3 to suggest several vacation destinations, we often receive variations of the same few ideas—a phenomenon known as *mode collapse* Hamilton (2024).

This lack of diversity in language model outputs represents a significant limitation. Today's aligned language models tend to produce lower entropy distributions than earlier generations of models (Zhang et al., 2024b), and when asked to generate (using random sampling) several responses to an open-ended prompt, the responses will often contain substantial near-duplicates (O'Mahony et al., 2024). This tendency can harm the utility of these models for subjective tasks where different users may have diverging preferences and needs (Zhang et al., 2024a). Sorensen et al. (2024) refer to the inability of today's LLMs to produce diverse generations as a failure of *pluralistic alignment*, which can lead to less useful and customizable AI systems.

While today's LLMs are evaluated at great length for knowledge and reasoning abilities, they are rarely evaluated for response diversity, and there are no widely-adopted benchmarks for assessing this trait. The majority of existing evaluation benchmarks are "mode-seeking", in the sense that they only assess the quality of the most likely generation of language

---

[*]Correspondence to <yimingz3@cs.cmu.edu>.
[1]Code and data are available at https://novelty-bench.github.io.

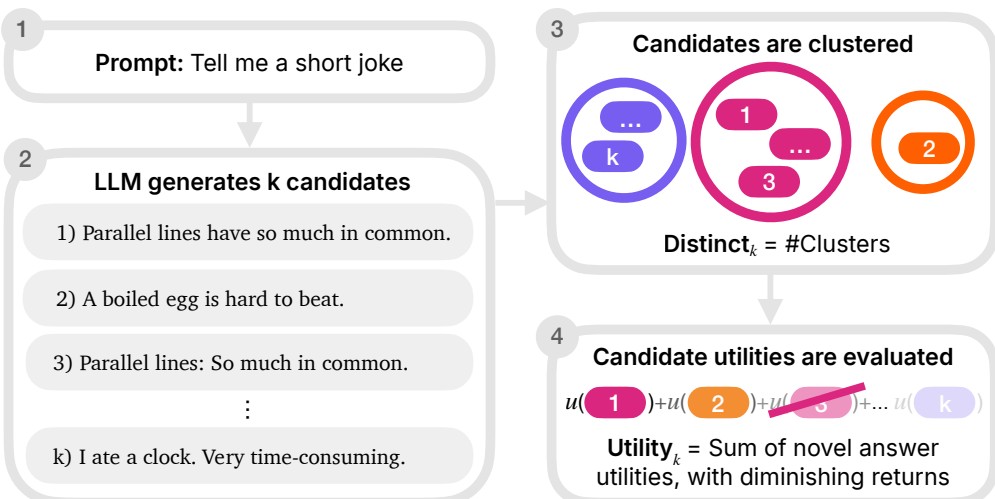

Figure 1: NOVELTYBENCH overview. We partition generations into functional equivalence classes and introduce two metrics: distinct$_k$ counts unique equivalence classes among $k$ samples, and utility$_k$ combines novelty and quality, weighing utility of individual generations by user patience ($p = 0.8$) and only considering *novel* generations.

models and do not assess the capability of models to produce meaningful alternatives.[2] This evaluation paradigm is problematic because it creates misaligned incentives: model developers focus on improving the quality of the single most likely generation rather than the diversity of the entire distribution of possible outputs.

In this work, we set out to measure not only what language models can generate, but also what they *cannot*. To this end, we propose NOVELTYBENCH (Figure 1), a benchmark for measuring how well language models can generate multiple differing but still correct answers to user requests that involve subjectivity, randomness, and creativity—in other words, queries that a room of humans would produce a large variety of answers to. We intend for our benchmark to serve as a complement to existing quality-based evaluation benchmarks, encouraging model developers to strive toward more pluralistic models, while also taking generation quality into account.

In summary, we introduce NOVELTYBENCH, a benchmark consisting of 1,100 prompts that we expect to elicit variable answers from humans, 100 of which are annotated with responses from eight human annotators. We propose a unified measure of novelty and quality which can be used to gauge how well an LLM can produce diverse, high-quality responses to a given prompt. We evaluate NOVELTYBENCH on twenty frontier models, including GPT-4o, Claude 3.5 Sonnet and Gemini 2.0 Pro, and find that all suffer from a lack of diversity. Our evaluation further reveals that larger and more capable models in the same model family tend to produce less diverse outputs, highlighting a concerning trend in the development of state-of-the-art language models. These findings underscore the need for future work on training techniques that promote diversity alongside quality, and for evaluation paradigms that better reflect the full spectrum of human preferences and creative capabilities.

## 2   Related Work

Prior work has established that language model outputs often exhibit biases towards specific demographic groups (Santurkar et al., 2023; Sun et al., 2023), genders (del Arco et al., 2024), countries (Durmus et al., 2023), or personas (Giorgi et al., 2024; Oketunji et al., 2023). This homogenization can limit LLM utility on downstream tasks (Agarwal et al., 2024;

---

[2]In an analysis of 67 benchmark papers published at COLM 2024 and ICLR 2025, we found that over 90% of these papers evaluate language models based on a single/best generation, instead of considering the full model output distribution.

Song et al., 2024). Research into diversifying language model outputs has seen several promising directions. Common approaches include adjusting decoding methods, such as increasing temperature (Shur-Ofry et al., 2024; Peeperkorn et al., 2024), or modifying training objectives (Li et al., 2016; Zhang et al., 2024b; Li et al., 2024; Lanchantin et al., 2025).

The idea that we should evaluate language models not only on the quality of their generations but also on their diversity is not entirely new. The 2020 Duolingo Shared Task involved generating several diverse but still correct translations of an input sentence (Mayhew et al., 2020). Recent work by Cheng et al. (2024) further highlights the probabilistic nature of commonsense knowledge in language models, and proposed an probabilistic evaluation method for commonsense knowledge that relies on open-ended generation instead of multiple-choice questions. Numerous studies of NLG systems have proposed metrics for evaluating diversity, based on perplexity (Lewis et al., 2017; Fan et al., 2018), n-gram overlaps (Zhu et al., 2018; Shu et al., 2019; Dušek et al., 2020), and embedding distances (Du & Black, 2019). However, existing metrics often correlate poorly with human judgments of content diversity (Tevet & Berant, 2021). In contrast to prior work, NOVELTYBENCH uniquely evaluates an LLM's capacity to generate a set of outputs that are simultaneously diverse and high-quality.

## 3 Benchmark

Our benchmark evaluates language model diversity using two distinct datasets: NB-CURATED which contains 100 prompts manually curated by the authors, and NB-WILDCHAT which consists of 1,000 prompts automatically curated from real user interactions with ChatGPT. LLM performance on the benchmark is evaluated based on the quality and novelty of the set of generations produced by the LLM.

### 3.1 Dataset Curation

To test language models' capabilities to produce novel generations, we need a dataset of prompts that elicit diverse responses. We developed two sets of such prompts: NB-CURATED contains prompts designed by the authors to have multiple valid answers, while NB-WILDCHAT is a collection of 1,000 prompts sourced from the WildChat-1M dataset (Zhao et al., 2023).

**NB-CURATED.** The curated dataset is designed to contain four distinct categories where diversity is expected.

- **Randomness**: prompts that involve randomizing over a set of options.
  Example: `Roll a make-believe 20-sided die.`
- **Factual Knowledge**: prompts that request underspecified factual information, which allow many valid answers.
  Example: `List a capital city in Africa.`
- **Creative Writing**: prompts that involve generating a creative form of text, including poetry, and story-writing.
  Example: `Tell me a riddle.`
- **Subjectivity**: prompts that request subjective answers or opinions.
  Example: `What's the best car to get in 2023?`

The authors of the paper then write responses to each of the prompts, resulting in 8 human responses per prompt. Since this is a rather homogeneous set of human annotators (all work/study at the same institution), we expect this response set to represent an approximate *lower* bound on actual human diversity. However, as we will show in Section 4.3, even this group produced more diverse responses than most state-of-the-art language models.

**NB-WILDCHAT.** WildChat is a collection of 1M real user conversations with ChatGPT, which is a valuable resource for understanding *real user needs of creativity* since they represent usage of language models in the wild. However, we cannot use this dataset directly because

the vast majority of its prompts do not allow for multiple valid answers.During processing, we first used Llama Guard 3 (Chi et al., 2024) to filter out inappropriate user prompts. A small number of IPs contributed a disproportionately high number of interactions, leading to topic bias (e.g., towards anime); we deduplicated the dataset based on user IP to achieve a more even topic distribution.After preprocessing, we used GPT-4o as a classifier to select prompts expected to elicit diverse generations.[3] To make sure that the selected prompts match the desired criteria, we manually labeled 100 prompts from the dataset (with 50 accepted and 50 rejected by GPT-4o), and found an 85% agreement between the classifier and the human labels. Randomly sampled prompts from this dataset are provided in Appendix A.3.

## 3.2 Evaluating Novelty

Models with diversity alone would not be sufficient for users: each of the individual generations should also be of high quality, with the ultimate goal of maximizing the cumulative utility of a user who interacts with the model. Thus, our evaluation framework aims to assess the extent to which the set of responses is mutually diverse and simultaneously high-quality, and our key metric directly models the cumulative utility for a user who observes a sequence of generations from a language model.

**Partitioning the output space.**   Defining diversity with automatic metrics is challenging due to the complex nature of language. Traditional metrics like n-gram overlap or embedding similarity often fail to capture meaningful diversity, as they may either overemphasize trivial differences (e.g., paraphrasing) or miss important semantic distinctions. Prior work has attempted to measure diversity using lexical variation (Li et al., 2016), embedding distances (Zhang et al., 2019), or entropy-based measures (Dušek et al., 2020), but these approaches typically struggle to distinguish between functional diversity and surface-level variations that provide little additional value to users (Tevet & Berant, 2021).

Accordingly, to understand the diversity of the output space of language models, we propose a method that learns to partition the output space into equivalence classes from human annotations. Each class represents one unique generation that is roughly equivalent to the others in the same class and different from the generations in other classes.

Partitioning the output space boils down to defining a notion of *equivalence* between pairs of generations. Language model outputs can be trivially different from each other (e.g., in terms of style, tone, or length). We argue that true diversity should go beyond these surface-level differences, since such variations provide little utility to the user. Therefore, we mainly consider *functional equivalence* in the context of this work, which defines two generations to be different if and only if a user who has seen one generation would likely benefit from seeing the other. For example, in a story generation task, if two generations are retelling the original story with identical plot, but different character names, then we consider them to be functionally equivalent.

While this definition of functional equivalence is intuitive for human evaluators, it is non-trivial to capture this notion of equivalence using existing similarity metrics.[4] To this end, the authors annotated 1,100 pairs of generations conditioned on prompts from both the NB-CURATED and NB-WILDCHAT datasets sampled from a diverse set of models (see Table 1 for a list of models). For annotation, we developed guidelines that focused on identifying whether two generations would provide distinct value to a user, rather than just surface-level differences.

From these annotated pairs, we used 1,000 for training and fine-tuned a `deberta-v3-large` model (He et al., 2021) to predict binary functional equivalence between two generations. To ensure that the classifier matches human judgment, we evaluated it on a held-out test set

---

[3]Prompt selection criteria are provided in Appendix A.2.

[4]We experimented with existing metrics such as BLEU (Papineni et al., 2002), ROUGE (Lin, 2004) and BERTScore (Zhang et al., 2019), but found their predictions to diverge from human judgment. See results in Appendix A.4.

of 100 pairs. The classifier achieved 79% accuracy compared to human labels, with an F1 score of 0.811, indicating strong performance on both positive and negative classes.

With the equivalence classifier, we can partition the output space into equivalence classes. For each new generation, we compare it against a random generation from each existing class. If the classifier determines it is functionally equivalent to any existing class, we assign it to the first such class found; otherwise, we create a new class with this generation. We define the $\text{distinct}_k$ metric as the number of equivalence classes in a partition of $k$ sample generations from a language model:

$$\text{distinct}_k := |\{c_i \mid i \in [k]\}| \tag{1}$$

where $c_i$ is the equivalence class that the $i$-th generation belongs to. For a sufficiently large $k$, $\text{distinct}_k$ quantifies the number of meaningfully unique generations that a user could have observed by sampling from the model $k$ times. A higher score indicates that the model is better at producing meaningful alternatives to the most likely generation.

**Unified measure of diversity and quality.**   How do we measure the cumulative utility of a sequence of generations $s_1, s_2, \ldots, s_k$? To achieve this, we first need a model of user behavior that describes how users interact with and consume language model generations. Inspired by the user model underlying the rank-biased precision metric in information retrieval (Moffat & Zobel, 2008), we assume that the user has a patience level $p \in [0, 1]$: after observing each additional generation, they have a probability $p$ of requesting an additional generation from the language model and observing the next generation, and a probability $1 - p$ of stopping interacting with the model. In other words, each additional generation should have a geometrically decaying effect on utility since it is more likely that the user simply would not have observed it.

To model the marginal utility of a distinct generation, we assume that the utility of a generation that is functionally equivalent to a previous generation is zero. Under our user model with patience $p$, the cumulative utility of a sequence of generations is given by:

$$\text{utility}_k := \frac{1 - p}{1 - p^k} \sum_{i=1}^{k} p^{i-1} \cdot \underbrace{\mathbb{1}\left[c_i \neq c_j, \forall j < i\right] \cdot u_i}_{\text{marginal utility of generation } i} \tag{2}$$

where $c_i$ is the equivalence class of the $i$-th generation, $u_i$ is the utility of the $i$-th generation given by some utility function, and $\frac{1-p}{1-p^k}$ is a normalization factor. In our evaluation setup, we set the patience level to 0.8. Note that as $p \to 0$, the metric converges to quality evaluation of a single generation, the de facto standard in language model evaluation.

**Computing utility.**   We use the `Skywork-Reward-Gemma-2-27B-v0.2` model (Liu et al., 2024) to score the quality of individual generations. We chose this reward model because it is one of the top performing models on RewardBench (Lambert et al., 2024), and outperforms larger models such as Nvidia's 70B Nemotron model (Wang et al., 2024).

The reward scores produced by the reward model are real-valued and difficult to interpret for humans. To map these scores into a more interpretable utility measure that outputs values in $\{1, 2, \ldots, 10\}$, we computed the distribution of reward values for 2,400 model generations on MT-Bench (Zheng et al., 2023) and used quality scores judged by GPT-4 to calibrate the reward mapping to match the score distributions.

## 4   Evaluation Results on NOVELTYBENCH

### 4.1   Models considered

We are particularly interested in the behavior of "aligned" language models which have been adapted using supervised finetuning and/or reinforcement learning to be conversational, instruction-following, and helpful. In particular, we consider a list of 20 state-of-the-art

| Model Provider | Variants | Distinct | Utility |
|---|---|---|---|
| Anthropic (Anthropic, 2024) | Claude-3.5 Haiku | 1.94 | 2.50 |
| | Claude-3.5 Sonnet | 1.76 | 2.36 |
| | Claude-3 Opus | 2.04 | 2.67 |
| OpenAI (OpenAI, 2024) | gpt-4o-mini | 2.65 | 3.11 |
| | gpt-4o | 2.88 | 3.27 |
| Gemini (Google, 2024) | gemini-1.5-pro | 1.85 | 2.73 |
| | gemini-2.0-flash-lite | 2.83 | 3.20 |
| | gemini-2.0-flash | 2.81 | 3.17 |
| | gemini-2.0-pro | 2.25 | 2.64 |
| Cohere (Cohere, 2024) | command-r7b | 3.58 | 3.35 |
| | command-r | 2.68 | 2.98 |
| | command-r-plus | 2.79 | 3.08 |
| Gemma 2 (Gemma Team et al., 2024) | gemma-2-2b-it | 5.66 | 4.63 |
| | gemma-2-9b-it | 3.25 | 3.93 |
| | gemma-2-27b-it | 3.03 | 3.77 |
| Llama 3 (Llama Team et al., 2024) | Llama-3.2-1B | 6.74 | 2.81 |
| | Llama-3.2-3B | 5.10 | 3.24 |
| | Llama-3.1-8B | 5.24 | 3.76 |
| | Llama-3.3-70B | 2.49 | 2.87 |
| | Llama-3.1-405B | 3.20 | 3.39 |

Table 1: List of models used as baselines for our benchmark, with their average number of distinct generations and cumulative utility scores out of 10 generations.

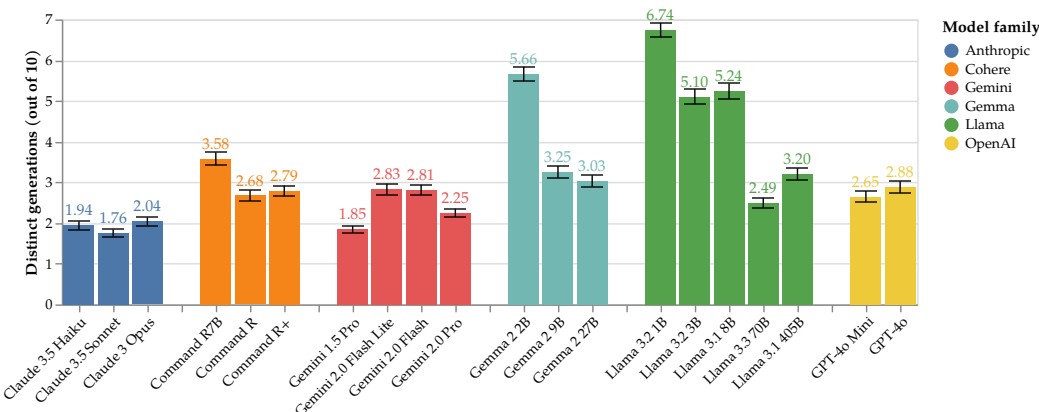

Figure 2: Average number of unique generations out of a sample of 10 for all prompts in NOVELTYBENCH. Error bars indicate 95% confidence intervals.

models from Anthropic, OpenAI, Google and Meta (listed in Table 1) as a representative sample of the performance of frontier models.

Where applicable, we disable retrieval-augmented generation and tool-use. While these augmentations undoubtedly influence language model diversity, examining their effects falls outside our scope. Instead, we focus our ablations on how various prompting techniques affect output diversity (Section 4.3). We evaluate model output distributions by independently sampling 10 generations with a temperature of 1—representing the best-case scenario for diversity, as most APIs default to lower temperatures or nucleus sampling.

## 4.2 State-of-the-art models struggle to generate diverse and useful outputs

Our evaluation reveals that frontier language models struggle to produce simultaneously diverse and useful responses. As shown in Figure 2, models like Claude 3 and GPT-4o

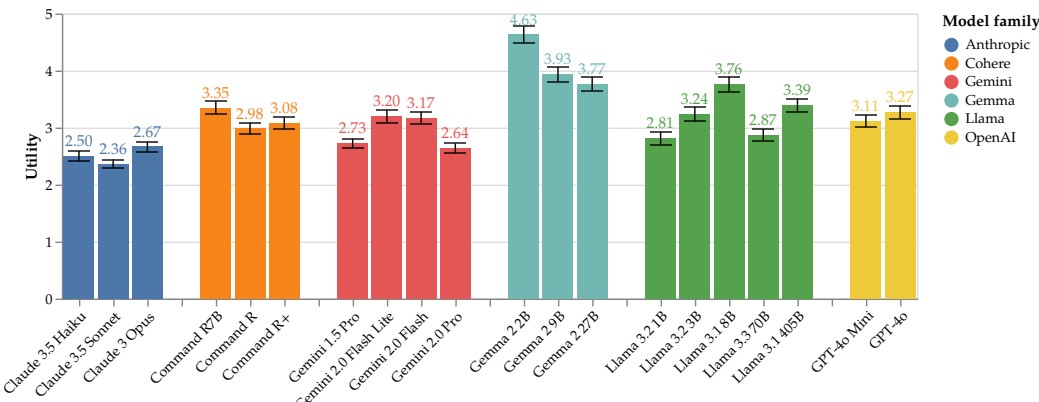

Figure 3: Cumulative utility of generations of state-of-the-art models on NOVELTYBENCH. A perfectly diverse and helpful model would have cumulative utility of 10. Error bars indicate 95% confidence intervals.

produce on average fewer than 3 distinct responses in 10 queries. Notably, smaller models such as Gemma 2-2B and Llama 3.2-1B demonstrate the highest diversity on our benchmark. Figure 3 further illustrates this issue by measuring cumulative utility of generations. This metric captures both overall response diversity and individual generation quality. In our utility evaluation, closed-source models such as Claude 3, Gemini, and GPT-4o all scored below 4 out of a maximum utility score of 10, further indicating that even frontier models fall short of providing diverse and useful responses.

While larger models typically excel in standard benchmarks, they demonstrate worse diversity in their generations compared to smaller models. In Figure 4, we show how the cumulative utility of model generations changes as a function of the patience parameter in our evaluation setup. We focus exclusively on open-weight models (Gemma 2 and Llama 3), for which the number of parameters is publicly known. Again, at patience=0, we are essentially evaluating just a single generation regardless of the diversity of the output distribution. As patience increases, additional novel generations are given higher weights. Unsurprisingly, larger models perform better when only the best generation is considered. However, as users demand more diversity from the output distributions of these models, the larger models degrade in utility more quickly due to their lack of diversity compared to their smaller counterparts. Our results here suggest that *progress on existing benchmarks that only consider a single output of the model can be misleading*: larger models may appear superior when evaluated on single-generation tasks, but they can underperform smaller models when users require diverse, creative solutions across multiple interactions.

## 4.3 Alternate prompting methods for inducing diversity

In practice, one way to deal with the lack of diversity in model output distributions is by employing alternative prompting strategies. For example, a user may paraphrase the prompt, or request a different answer from the model if they are not satisfied with the current output. In this section, we explore and compare the following methods to solicit more diverse responses from the models:

- **Resampling:** The default approach where we generate multiple independent samples from the model using the same prompt with temperature set to 1.0. Each generation is independent and the model has no knowledge of previous generations (same setting as in Section 4.2).

- **Paraphrasing:** For each generation, we create a distinct paraphrase of the original prompt, then use these varied prompts to elicit different responses.

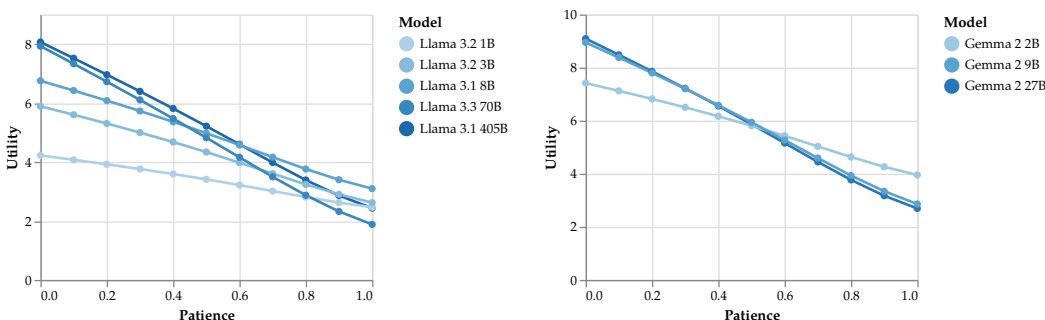

Figure 4: As users demand more diverse generations, models become less useful. Larger models suffer more from degradation in utility.

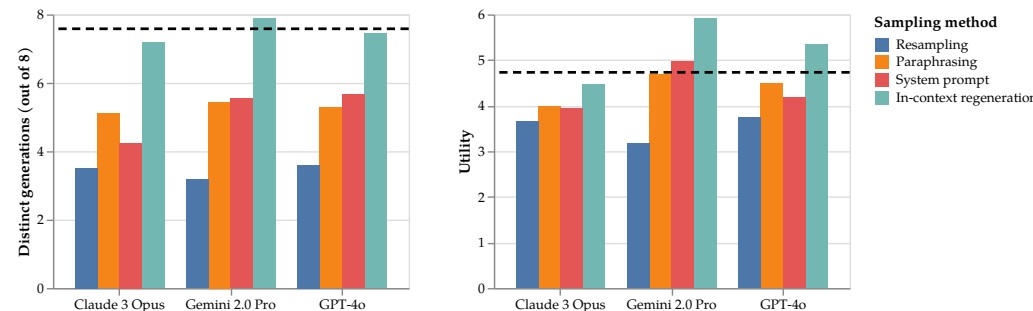

Figure 5: Alternative prompting methods *can lead to improved novelty.* The dashed lines report diversity and utility of answers handwritten by authors.

- **System prompt:** We prepend the system prompt "You are a producer of unique answers, and you strive to tell each user a novel answer to their question" to encourage the model to prioritize diversity in its responses.

- **In-context regeneration:** After each generation, we explicitly ask the model to provide a different answer while keeping all previous answers in the conversation context. This allows the model to see its previous responses and deliberately generate something new.

To establish a human baseline, 8 authors of this paper answered each question in the NB-CURATED dataset, providing a reference point for the diversity achievable by a group of humans. On a subset of 100 curated prompts, we evaluated the performance of three state-of-the-art models (Claude-3 Opus, GPT-4o and Gemini 2.0 Pro) similarly over 8 generations.

As shown in Figure 5, these alternative prompting strategies demonstrate varying degrees of effectiveness. Paraphrasing and system prompting are only marginally effective—indicating that asking a model for "creative outputs" does not work very well. In-context regeneration is the most successful approach, with all three models roughly matching the diversity of human writers (the dashed lines in figures). This suggests that state-of-the-art LLMs can indeed generate diverse answers when explicitly constrained by their previous outputs in context. Under this strategy, GPT-4o and Gemini 2.0 Pro even surpass the cumulative utility scores of human writers due to the added diversity. While our findings indicate that prompt engineering could partially address diversity limitations, they also reveal that this diversity is not inherently built into the models' output distributions. Rather, it must be deliberately elicited through specific prompting techniques.

| Suggest a name for a dappled-gray filly living in the mountains. | Name one person who won the Fields Medal. | Roll a make-believe 20-sided die. | What's the best car to get in 2023? |
|---|---|---|---|
| How about the name **Silver Mist**? It evokes the filly's dappled-gray coat *[...]* | One person who has won the Fields Medal is **Maryam Mirzakhani**. She received *[...]* | Alright! Rolling a 20-sided die... *[...]* You rolled a **14**! | If you're looking for a single, well-rounded option in 2023, the **Toyota RAV4 Hybrid** is *[...]* |
| How about the name **"Silvermist"**? It captures the ethereal beauty of her dappled-gray coat *[...]* | One person who won the Fields Medal is **Terence Tao**, who received *[...]* | Alright! Rolling the imaginary 20-sided die... *[...]* You rolled a **17**! | If I had to give you just one suggestion for the best car to get in 2023, it would be the **Toyota RAV4 Hybrid**. *[...]* |
| How about **"Silvermist"**? It evokes the ethereal beauty of her dappled-gray coat *[...]* | One person who won the Fields Medal is Maryam Mirzakhani. She received *[...]* | Sure! Rolling a 20-sided die... *[...]* You rolled a **15**! | The **2023 Toyota RAV4 Hybrid** is an excellent choice. *[...]* |
| How about **"Luna Cascade"**? It evokes the soft, silvery hues of her dappled coat *[...]* | One person who won the Fields Medal is **Maryam Mirzakhani** in 2014. She was *[...]* | Sure! Rolling a make-believe 20-sided die for you... *[...]* You rolled: **17** | Alright, if I had to give you just *one* recommendation: **Toyota RAV4 Hybrid**. *[...]* |
| Sure! How about **Luna Mist**? It's elegant, evokes the image of moonlight *[...]* | Sure! One person who won the Fields Medal is **Maryam Mirzakhani** in 2014. She was *[...]* | Sure! *Rolls a make-believe 20-sided die* ... You got a **13**! | The **Toyota RAV4 Hybrid** is an excellent all-around choice for 2023. *[...]* |

Table 2: GPT-4o responses to NB-CURATED prompts. Even with full random sampling, GPT-4o fails to produce diverse outputs. Bold/italics preserved from source outputs.

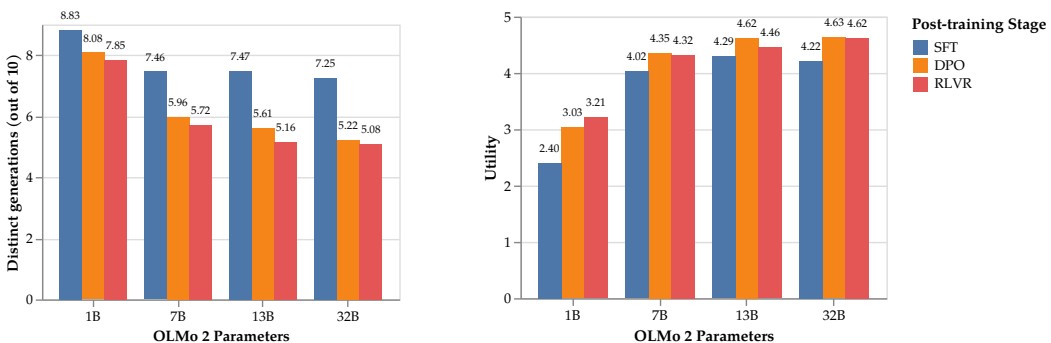

Figure 6: Diversity and utility analysis of OLMo 2 models across alignment stages.

## 4.4 How does alignment affect diversity?

The recent OLMo 2 series of models (OLMo et al., 2025), with weights released after each post-training stage, provides an opportunity to investigate the mechanistic effects of alignment on output diversity. We evaluate OLMo 2 models at all sizes (1B, 7B, 13B, 32B) after three stages of post-training: supervised fine-tuning (SFT), direct preference optimization (DPO), and finally reinforcement learning with verifiable rewards (RLVR) to analyze the effect of these three post-training stages on output diversity in Figure 6.

Across model sizes, we observe that each alignment stage progressively reduces model diversity, with significant drops occurring during DPO. This finding suggests that current alignment procedures inadvertently suppress distributional diversity. Interestingly, the improvement in generation quality compensates for the loss in diversity. Specifically, we observe utility gains when transitioning from SFT to DPO, suggesting that while post-training algorithms may further reduce diversity, they can improve the overall usefulness

of the remaining diverse outputs, highlighting a complex trade-off between alignment objectives and output diversity.

Our results suggest that alignment algorithms should be designed with output diversity in mind. Future work should investigate how to strike a better balance in alignment algorithms to achieve both safety and helpfulness objectives while maintaining the rich diversity that makes language models useful for creative tasks.

## 5    Discussion and Future Work

Our findings suggest that language models lack *distributional* diversity; given a fixed prompt, their outputs are heavily skewed toward certain responses over other equally valid ones. Users can to some extent elicit novel generations by modifying their prompt (for example, adding more detailed instructions) or else iteratively asking "give me a different answer" in a multi-turn fashion. However, these workarounds are cumbersome and don't address the fundamental issue that modern language models output overly low-entropy distributions which limit their generative capabilities. Methods that directly improve distributional diversity remain valuable as they will ensure diverse outputs across all generation settings, regardless of prompting strategy.

One key question our benchmark does not address is the extent to which users want diversity. For some prompts, like the dice-rolling one in Table 2, the need for diverse generations is immediately obvious. However, for other prompts, like the car recommendation example in Table 2, it may in fact be desirable for the LLM to give a consistent response each time. Should a LLM's generations represent the opinions of a diverse group of hypothetical humans, or should they possess fixed, consistent behavior? To take the "best car to buy in 2023" example, any one human will likely repeat the same answer when asked over and over again, but across a population of humans, many different cars will be suggested.[5] Consistency in LLM outputs is crucial for high-stakes applications like healthcare or legal assistance that rely on reliably correct answers, but it can conversely lead to the LLM imposing a single view on users.

Our findings on distributional diversity suggest that current models struggle with both objectives–they neither fully represent the diversity of valid human responses nor maintain perfect consistency across similar prompts. Future work should explore how to balance these competing goals, potentially by developing models that can explicitly modulate between diversity and consistency based on specific task requirements and user expectations. Furthermore, we would like to see user studies that investigate user interaction patterns with LLMs, how expectations of consistency/diversity vary across categories of prompts, and how users' methods of interacting with LLM services affect perceived creativity. These findings will be useful both in guiding the behavior of future generations of language models and in developing evaluation metrics that better reflect real-world usage.

While there is starting to be some research in diversity-promoting post-training techniques, that is an area that deserves more attention from researchers. Maximum likelihood estimation inherently optimizes language models toward a single response for each prompt. This, combined with alignment algorithms which often associate a prompt with a single high-reward response, unsurprisingly causes language models to favor a certain response among several reasonable choices (Zhang et al., 2024a). Recent work has begun addressing this limitation through modified training objectives during supervised fine-tuning (Li et al., 2024) and preference optimization (Lanchantin et al., 2025), but these approaches represent only initial steps. Future work should continue exploring a broader range of methods that improve distributional diversity of language models while maintaining response quality.

---

[5]Human responses to the prompts in Table 2 can be found in the Appendix.

## Acknowledgements

This research was supported by an OpenAI Superalignment Fellowship (YZ) and sponsorship from Cisco (YZ, DI, BW). We also thank Google and Cohere for providing access to their models through API credits.

## Contributions

**YZ** came up with the idea of evaluating language models beyond their most likely outputs, and was responsible for running experiments.

**HD, HL and VS** contributed to the codebase of NOVELTYBENCH.

**XL** set up the website for the benchmark.

**BW** contributed to the collection of the NB-CURATED dataset, and was responsible for visualizations in the paper.

**SH** was responsible for annotating data for training and evaluating the equivalence classifier, and verified prompts in the NB-WILDCHAT subset.

**DI** supervised the project and suggested evaluating model creativity against humans, and proposed alternative prompting methods for inducing diversity from models.

**All authors** provided contributions to designing experiments, data annotation, and paper writing.

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

# A  Additional Evaluation Details

## A.1  Diversity of Reasoning Models

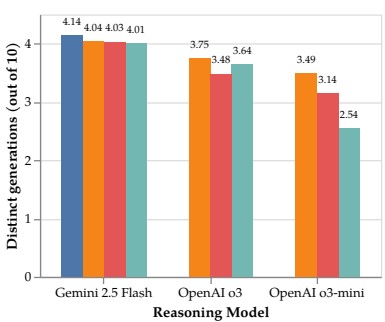 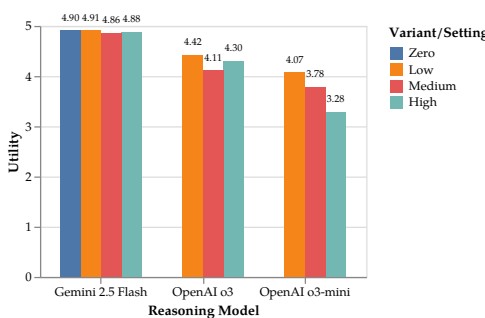

Figure 7: Diversity and utility performance of reasoning models on NoveltyBench. OpenAI's models demonstrate a trend where increased reasoning effort leads to reduced diversity and utility.

Our core analysis focuses on the 20 most representative (non-reasoning) language models from major providers to establish a comprehensive baseline across model families and sizes, since reasoning capabilities are not yet widely available across model families at the time of writing. That said, as more frontier models carry out an explicit reasoning process before responding to user queries, we believe it would be interesting to see how such reasoning affects diversity.

To this end, we evaluate OpenAI and Google's latest reasoning models (o3, o3-mini, and Gemini-2.5-flash) for which inference-time reasoning effort can be explicitly set, and explored how they impact diversity and utility on NoveltyBench. For the Gemini model, setting the reasoning budget had minimal effect on our evaluation. On the other hand, we see a trend with OpenAI's models where additional reasoning seems to lead to less diverse and less useful outputs. We note that although reasoning models are usually used to solve problems with a single correct answer (e.g., math and coding), preserving diversity is still critical for the effectiveness of inference-time scaling strategies such as pass@k. Future work should further investigate how generation diversity affects creativity in problem solving, and its implications on inference-time scaling.

## A.2  Prompt for Selecting Diverse Prompts

The following prompt was used with GPT-4o to select prompts from the WildChat dataset that allow for diverse responses:

## A.3  Example Prompts from NB-WILDCHAT

The following table presents a sample of prompts from the NB-WILDCHAT dataset that were selected using the GPT-4o classifier:

## A.4  Evaluation of equivalence classification methods

To determine the equivalence of two language model generations, we used a variety of existing text similarity metrics such as BERTScore, BLEU, ROUGE, as well as fine-tuning OpenAI models (`GPT-3.5-turbo` and `GPT-4o-mini`) and a pre-trained DeBERTa model (`microsoft/deberta-v3-large`) on a manually labeled dataset of 1,000 pairs of LM-generated text.

On a validation set of 100 pairs, we found that a DeBERTa reached the highest AUC score of 0.81, outperforming both fine-tuned OpenAI models and heuristic-based methods

You are helping select prompts for a benchmark that measures language models'
ability to generate diverse, high-quality alternative answers. For a prompt
to qualify, it should:
1. Allow diverse responses: The prompt should enable multiple valid distinct
responses. For example, a prompt that asks for a salmon recipe, a chess
move in a given position, or a continuation of a story would allow diverse
responses. In contrast, a prompt that asks for a specific fact, or a rewrite
of input text would not.
2. Both the prompt, and the desired response should be in English.
3. Request a natural language response. Requests that ask for code, images
or other output formats do not qualify.
4. Make a single clearly interpretable request. For example, "recommend a
reliable espresso machine" is clear, while "espresso machine" is not.

Classify the following prompt based on these criteria, and format the provided
prompt. Output using the provided JSON format.

Table 3: System prompt used for selecting diverse prompts from the WildChat dataset

**Example Prompts**

Describe me as a smart, funny, and kind guy who could change anybody's life
for an "about me" section on a dating app. Make it professional and smart.

Write an example illustrating how climate change can be considered a natural
phenomenon rather than man-made.

How can I make a collage more interesting?

Value yourself as a star, sun, and a tall man.

Can you introduce "audit" as a research method in social science?

Write a story about a wizard being sent back in time to when he was born after
dying before mastering all types of magic.

Write a background of study on the topic "Navigating the Role of Campus
Directors as Managers: A Narrative Inquiry of Their Lived Experiences."

How can I learn to code?

Write a story about a man who uses a magical remote to turn himself into his
dream girlfriend, changing himself one step at a time.

What are the best free VPNs? List the reasons and reviews.

Table 4: Randomly sampled prompts from the NB-WILDCHAT dataset

significantly. After choosing an appropriate confidence interval, the fine-tuned DeBERTa model has an accuracy of 71.0% and an F1 score of 0.811.

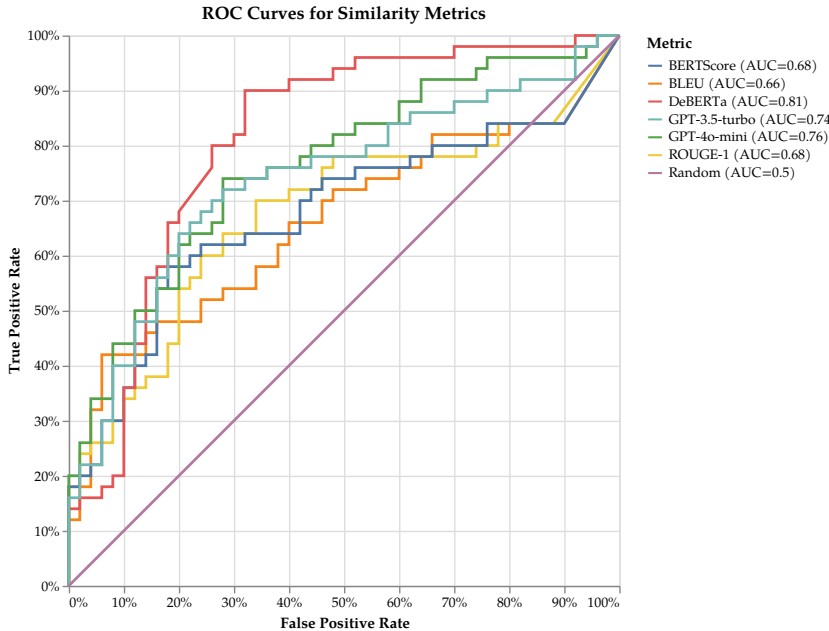

Figure 8: ROC curve for various similarity classification methods.

## A.5 Human Annotations

Eight human annotators, drawn from members of our institution, responded to each prompt in NB-CURATED. Examples of the human annotations corresponding to the prompts from Table 2 are presented in Table 5.

| Suggest a name for a dappled-gray filly living in the mountains. | Name one person who won the Fields Medal. | Roll a make-believe 20-sided die. | What's the best car to get in 2023? |
|---|---|---|---|
| Jumpy | Kunihiko Kodaira | 17 | Subaru BRZ |
| Maximus | Akshay Venkatesh | 18 | Honda Civic |
| Oolong | James Maynard | 9 | Toyota Camry |
| Winter | Gerd Faltings | 20 | 2025 Rivian R1S |
| Teazle | Maryam Mirzakhani | 7 | Prius |
| Tony | Maryna Viazovska | 20 | We decided on a Mazda CX5. It was the best value per dollar. |
| Monte | June Huh | 19 | Tesla Model Y |
| Greg | Maryam Mirzakhani | 13 | Volvo XC90 |

Table 5: Human responses to prompts from NB-CURATED.

