# OpenReview forum: "NoveltyBench: Evaluating Language Models for Humanlike Diversity"
_colmweb.org/COLM/2025/Conference — COLM 2025_

### Official Review · Reviewer_KaJt · 2025-04-28

**Rating:** 6
**Confidence:** 3
**Ethics Flag:** 1

**Summary:**

The paper introduces NOVELTYBENCH, a benchmark designed to assess the ability of language models to generate diverse and novel outputs. NOVELTYBENCH contains a set of 1,100 prompts curated to elicit varied answers, along with real-world user queries. NOVELTYBENCH contains two proposed metrics, Distinctk and Utilityk, evaluate the diversity and quality of generated content. The study evaluates 20 leading models, including GPT-4o, Claude 3.5 Sonnet and Gemini 2.0 Pro. They find that even state-of-the-art systems produce significantly less diversity compared to human responses, with larger models often being less diverse than their smaller counterparts. The research highlights a critical lack of distributional diversity in existing models, suggesting a need for new training and evaluation methods that prioritize both creativity and quality.

**Reasons To Accept:**

- The topic of diversity in LLM-generated content is intriguing. The diversity of LLM outputs is important. This work can help advance research in this area.
- The development process of NOVELTYBENCH is well-thought-out, with the proposed metrics, Distinctk and Utilityk, being particularly apt and logically constructed.
- Consequently, both the dataset and the proposed metrics hold significant potential utility for the field.
- The comprehensive evaluation of 20 leading LLMs, including detailed discussions of the findings, adds substantial value to this study.

**Reasons To Reject:**

- While the diversity of LLM-generated content is undoubtedly valuable in practical applications, one might question whether these models should inherently possess such diversity.

Human responses vary due to individual life experiences and perspectives, whereas an LLM, with its fixed architecture and parameter set, functions more like a single person at a given point in time. Thus, it's expected that its outputs lack diversity.

Considering different LLMs, their training data tends to be quite similar. Simply shuffling this data without introducing meaningful orders may not significantly alter output diversity.

Hence, stating that "LLMs lack diversity" could be akin to saying "KBQA systems' outputs aren't flexible enough, since they are KB entities" which isn't necessarily a flaw of the LLMs but rather an issue with what we choose to evaluate.

This perspective somewhat diminishes the contribution of the paper.

- Separately reporting the performances across the four question categories (Randomness, Factual Knowledge, Creativity, and Subjectivity) as outlined in Section 3.1 could provide deeper insights.

- The evaluation of the two metrics, Distinctk and Utilityk, relies heavily on model performance. Reporting accuracies, such as "the classifier achieved 79% accuracy compared to human labels," doesn't convey high confidence. Selecting a smaller subset for comparison between model scores and human expert evaluations, alongside calculating confidence intervals, would give researchers clearer insight into the reliability of these metrics.

- The evaluation lacks secondary metrics, such as standard deviations for Distinctk and Utilityk, which could enhance the understanding of the evaluation results.

---

> ### Author Response · Authors · 2025-06-01
>
> We thank Reviewer KaJt for the detailed review and thoughtful engagement with our work. We appreciate the recognition that diversity in LLM content is "intriguing" and "important," that our benchmark development is "well-thought-out," and that our comprehensive evaluation "adds substantial value."
>
> **1. Should models be inherently diverse?**
>
> The reviewer raises a fundamental question: "whether these models should inherently possess such diversity" given that an LLM "functions more like a single person at a given point in time." However, we argue that there *are* requests users regularly make to LLMs for which users expect different answers to be produced every time. User expectations of a language model and a KBQA system are meaningfully different:
>
> - When users ask "Tell me a joke" or "Suggest a vacation destination," they often want variety across interactions. Unlike KBQA systems that should always return factually correct entities, users making these creative/subjective queries benefit from multiple valid perspectives—chatbots would not be fulfilling user demands if they tell the same joke or recommend the same vacation spots repeatedly.
>
> - While a single human asked the same question repeatedly might give consistent answers, LLMs serve millions of users simultaneously—they should be capable of reflecting the diversity of human responses across populations. In this work, we are interested in the scenario of multiple people each asking the same question to the LLM system, rather than one person repeatedly asking the same question multiple times.
>
> Building on recent work on *pluralistic* alignment [1], we believe there is merit in building language models that are *distributionally diverse*. Nevertheless, as follow-up research, we plan to design surveys that explore user expectations of LLM diversity and answer whether distributional diversity is actually what users want.
>
> [1] [A Roadmap to Pluralistic Alignment](https://arxiv.org/abs/2402.05070)
>
> **2. Separate reporting across question categories**
>
> We have additionally reported results across the four categories (Randomness, Factual Knowledge, Creativity, and Subjectivity). Overall, models generally perform worse at providing underspecified factual information and slightly better at other categories like randomness. However, in none of the four categories do models provide reasonable levels of diversity and utility. We will incorporate these results in our revision.
>
> [Diversity results](https://ibb.co/d4NKV8tD) | [Utility results](https://ibb.co/LdMVHdmF)
>
> **3. Metric reliability**
>
> You are correct that evaluation of the metrics relies heavily on model performance, and this is a limitation of our work. In an ideal setting, both diversity and quality of model responses would be evaluated with human annotators, but it is near-impossible to build a standardized benchmark that relies entirely on human evaluation. We see our benchmark as a first step that model trainers should take to assess their models' diversity capabilities. When possible, automatic evaluation of diversity should be followed up with user studies and other human evaluation.
>
> To address the 79% classifier accuracy concern: while not perfect, this substantially outperforms existing similarity metrics (BLEU: 66% AUC, BERTScore: 68% AUC vs. our DeBERTa: 81% AUC, as shown in Appendix A.3). We acknowledge this limitation and suggest that future work could explore more sophisticated equivalence detection methods.
>
> **4. Statistical confidence**
>
> We have computed 95% confidence intervals for our main evaluations. Overall, the error margins are small because we evaluate over a large number of prompts (1,100 total). Here are the updated results with confidence intervals:
>
> | Model | Distinctness | Utility |
> |-------|-------------|---------|
> | Claude 3 Opus | 2.04 ± 0.10 | 2.67 ± 0.09 |
> | Claude 3.5 Haiku | 1.94 ± 0.10 | 2.50 ± 0.09 |
> | Claude 3.5 Sonnet | 1.76 ± 0.09 | 2.36 ± 0.08 |
> | Command R | 2.68 ± 0.13 | 2.98 ± 0.10 |
> | Command R+ | 2.79 ± 0.13 | 3.08 ± 0.10 |
> | Command R7B | 3.58 ± 0.16 | 3.35 ± 0.12 |
> | GPT-4o | 2.88 ± 0.15 | 3.27 ± 0.12 |
> | GPT-4o Mini | 2.65 ± 0.14 | 3.11 ± 0.11 |
> | Gemini 1.5 Pro | 1.85 ± 0.08 | 2.73 ± 0.08 |
> | Gemini 2.0 Flash | 2.81 ± 0.13 | 3.17 ± 0.11 |
> | Gemini 2.0 Flash Lite | 2.83 ± 0.13 | 3.20 ± 0.11 |
> | Gemini 2.0 Pro | 2.25 ± 0.10 | 2.64 ± 0.08 |
> | Gemma 2 27B | 3.03 ± 0.14 | 3.77 ± 0.12 |
> | Gemma 2 2B | 5.66 ± 0.18 | 4.63 ± 0.15 |
> | Gemma 2 9B | 3.25 ± 0.15 | 3.93 ± 0.13 |
> | Llama 3.1 405B | 3.20 ± 0.15 | 3.39 ± 0.12 |
> | Llama 3.1 8B | 5.24 ± 0.19 | 3.76 ± 0.13 |
> | Llama 3.2 1B | 6.74 ± 0.18 | 2.81 ± 0.12 |
> | Llama 3.2 3B | 5.10 ± 0.18 | 3.24 ± 0.13 |
> | Llama 3.3 70B | 2.49 ± 0.12 | 2.87 ± 0.10 |
>
> We will present appropriate statistical confidence measures in our revision to help readers better interpret the findings.

---

> ### Comment · Reviewer_KaJt · 2025-06-03
>
> The authors' comments have partially solved the problems, and I increase my rating from 5 to 6.

---

### Official Review · Reviewer_SHky · 2025-05-11

**Rating:** 6
**Confidence:** 3
**Ethics Flag:** 1

**Summary:**

This paper presents NoveltyBench, a new benchmark to systematically evaluate the diversity and creativity of language model outputs. 'To address this, NoveltyBench includes two datasets: NB-CURATED, with 100 author-designed prompts covering randomness, subjectivity, creativity, and underspecified factual questions; and NB-WILDCHAT, a filtered set of 1,000 diverse real-world prompts from ChatGPT logs. The authors define two evaluation metrics: (1) Distinct@k, measuring the number of functionally distinct generations; and (2) Utility@k, a novel, patience-weighted cumulative utility measure combining diversity and quality.

The benchmark evaluates 20 leading LLMs (GPT-4o, Claude 3, Gemini, Llama, etc.), and results reveal a surprising trend: larger models are often less diverse than smaller ones

**Questions To Authors:**

- For the LLaMA models evaluated, were the generations produced by the base models or the instruction-tuned variants?

- Have you considered evaluating diversity using lexical similarity metrics (e.g., n-gram overlap) or embedding-based measures (e.g., BERTScore, cosine distance)? How do these compare to your functional equivalence classifier?

- Could prompt complexity or ambiguity serve as an alternative explanation for variations in generation diversity, beyond model entropy? It would be interesting to explore this in future work.

**Reasons To Accept:**

- The studied problem is important: how to systematically evaluate creative diversity, not just correctness or helpfulness, in LLMs. The proposed benchmark is a critical addition to the evaluation landscape.

- The findings of the paper are interesting: larger models often exhibit lower diversity, and that prompting alone cannot fix the underlying entropy collapse.

- By implementing a fine-tuned equivalence classifier and reward model mapping, the paper provides a replicable, scalable evaluation tool for future model developers.

**Reasons To Reject:**

- The paper would benefit from a more fine-grained analysis of how different post-training stages (e.g., supervised fine-tuning, alignment) affect model diversity. For instance, analyzing intermediate checkpoints from models like Tulu v3 could offer deeper insight and strengthen the empirical findings.

-  The evaluation of this paper merely focus on chat prompts. This can be expanded with the reasoning/coding tasks.

- The utility metric relies on a third-party reward model (Skywork-Reward-Gemma-2-27B), which may introduce biases or artifacts based on how it was trained.

- Although NB-WILDCHAT is sourced from real user queries, the filtering and selection process (e.g., classifier-driven prompt curation and IP deduplication) may still underrepresent certain prompt types or domains (e.g., technical queries, multilingual prompts, dialog turns). It would be better to provide some additional discussions on this.

---

> ### Author Response · Authors · 2025-05-31
>
> We thank Reviewer SHky for the thoughtful and detailed review. We are glad that the reviewer agrees our work addresses an important problem and provides a "critical addition to the evaluation landscape." We appreciate the constructive feedback and address the concerns below:
>
> **1. Fine-grained analysis of post-training stages**
>
> This is an excellent suggestion that would provide valuable mechanistic insights into diversity degradation due to post-training. Following the reviewer's suggestion, we have added evaluations of OLMo 2 models (built upon Tulu 3) after three stages of post-training (SFT, DPO and RLVR), and analyzed diversity loss after each stage.
>
> [Diversity figure](https://ibb.co/LzGjGhCz)
>
> [Utility figure](https://ibb.co/4ZsQh6QM)
>
> Our analysis shows that each alignment stage progressively reduces model diversity, with the most significant drop occurring during supervised fine-tuning. This finding suggests that our hypothesis that current alignment procedures inadvertently suppress distributional diversity. Interestingly, in some cases, the improvement in generation quality makes up for the loss in diversity (utility gain from SFT -> DPO). Future work should investigate how to strike a balance in alignment algorithms to get the best of both worlds.
>
> **2. Expansion to reasoning/coding tasks**
>
> We appreciate this suggestion and acknowledge that our current benchmark focuses primarily on open-ended, creative tasks rather than reasoning or coding scenarios. We deliberately focused on tasks where diversity is *expected and desirable* - areas where multiple valid answers exist and where users would reasonably benefit from variety.
>
> However, we agree that understanding the impact of explicit reasoning on model diversity is valuable. We have added preliminary evaluations of reasoning models (o3, o3-mini, Gemini-2.5-flash) and found interesting patterns where additional reasoning effort can sometimes reduce diversity. We plan to explore this further in future work.
>
> **3. Third-party reward model bias**
>
> This is a valid methodological concern. We chose Skywork-Reward-Gemma-2-27B because it achieved top performance on RewardBench, outperforming other reward models including NVIDIA's Nemotron-70B model. While we acknowledge that any specific choice of reward model may introduce biases, the use of a standardized reward model is necessary for benchmark reproducibility and removes dependence on expensive manual annotations.
>
> In preliminary experiments, we also tested LLM-as-a-Judge (specifically GPT-4o), but found it produced skewed score distributions (most scores between 6-10), unlike a scalar reward distribution that is easier to calibrate as we did in our approach.
>
> **4. Dataset filtering and representation**
>
> We appreciate this concern about potential underrepresentation in NB-WILDCHAT. Our filtering process was designed to balance comprehensiveness with quality: we used Llama Guard 3 for safety filtering, IP-based deduplication to reduce topic bias, and GPT-4o classification to select prompts likely to elicit diverse responses (85% agreement with human judges).
>
> While this process may underrepresent certain domains (technical queries, multilingual prompts), we believe it provides a reasonable approximation of diversity-seeking user intents. We acknowledge this limitation and suggest that future work could develop domain-specific diversity benchmarks to complement our general-purpose evaluation.

---

> > ### Comment · Reviewer_SHky · 2025-06-10
> >
> > Thanks for your response. I will keep my current score for this paper.

---

> ### Author Response · Authors · 2025-05-31
> **Response to additional questions**
>
> **Q1: LLaMA model variants**
>
> **Answer**: We evaluated the **instruction-tuned variants** for all LLaMA models. We will clarify this explicitly in Table 1 in our revision.
>
> **Q2: Lexical similarity metrics comparison**
>
> **Answer**: Before developing our classifier, we extensively experimented with off-the-shelf similarity metrics, but found they performed poorly at capturing functional equivalence. For example, two stories with identical plots but different wording would be considered "diverse" by n-gram metrics but functionally equivalent to users.
>
> Appendix A.3 provides detailed comparisons. Our results show:
> - **BLEU**: AUC = 0.66
> - **ROUGE-1**: AUC = 0.68
> - **BERTScore**: AUC = 0.68
> - **Our DeBERTa classifier**: AUC = 0.81
>
> While our trained classifier is far from perfect, it does outperform existing metrics (e.g., lexical similarity-based ones).
>
> **Q3: Prompt complexity/ambiguity as alternative explanation**
>
> This is an insightful observation that deserves further investigation. In Section 4.3, we partially address this through experiments with prompt paraphrasing and diversity-seeking system prompts, neither of which significantly increased diversity. However, we did not systematically study how prompt complexity affects generation diversity.
>
> We agree this would be valuable for future work. That said, we argue that models with sufficient entropy should exhibit diversity across all reasonable prompts, regardless of specific wording - this is the type of robust distributional diversity that model developers should strive for.

---

### Official Review · Reviewer_3g5a · 2025-05-13

**Rating:** 7
**Confidence:** 4
**Ethics Flag:** 1

**Summary:**

This paper introduces NOVELTYBENCH, a benchmark designed to evaluate the capability of language models to generate diverse and novel outputs. It addresses the issue of “mode collapse,” where language models fail to produce varied responses. The benchmark consists of curated prompts designed to elicit diverse outputs and real-world user queries from WildChat. The study evaluates 20 advanced language models, finding that larger models often produce less diversity than smaller models in the same family. It also demonstrates that alternative prompting strategies like in-context regeneration can increase diversity, highlighting fundamental limitations in current models regarding generative diversity.

**Questions To Authors:**

See above

**Reasons To Accept:**

- Introduces NOVELTYBENCH, a benchmark measuring the diversity and novelty of language model outputs.
- Provides evidence that larger and more capable language models tend to produce less diverse outputs.
- Demonstrates alternative prompting methods (e.g., paraphrasing, system prompts, in-context regeneration) to enhance diversity.
- Suggests the need for new training and evaluation paradigms that prioritize diversity and creativity alongside quality

**Reasons To Reject:**

1. Lack of reasoning models as baselines for benchmarking purpose.
2. The temperature setting may influence the Distinct
3. need additional analysis of the diversity and distribution of dataset itself to avoid potential bias

---

> ### Author Response · Authors · 2025-05-31
>
> We thank Reviewer 3g5a for the thoughtful review and positive assessment of our work. We are pleased that the reviewer recognizes the importance of our benchmark in addressing the critical issue of mode collapse in language models. Below, we address the specific concerns raised:
>
> **1. Lack of reasoning models as baselines**
>
> We chose to focus on the 20 most representative (non-reasoning) language models from major providers to establish a comprehensive baseline across model families and sizes, since reasoning capabilities weren't generally available among model families at the time of writing. That said, we agree that as more models carry out an explicit reasoning process before responding to user queries, it would be interesting to see how such reasoning affects diversity.
>
> We have evaluated OpenAI and Google's latest reasoning models (o3, o3-mini, Gemini-2.5-flash) for which inference-time reasoning effort can be explicitly set, and explored how they impact diversity and utility on NoveltyBench ([diversity figure](https://ibb.co/kgw8vCfm) and [utility figure](https://ibb.co/TxX4GPNV)). For the Gemini model, setting the reasoning budget has minimal effect on our evaluation. On the other hand, there appears to be a trend with OpenAI's models where additional reasoning leads to less diverse (and useful) outputs. Future research should further investigate the effect of reasoning on generation diversity.
>
> **2. Temperature setting influence on diversity**
>
> This is an excellent point that deserves clarification. We deliberately chose temperature=1.0 to represent the **best-case scenario for diversity** as stated in Section 4.1 (L209). This choice was intentionally conservative—by using the most "diverse" setting, our results represent an upper bound on the actual diversity users experience at deployment time. A single choice is also somewhat inevitable to achieve consistent comparison among models.
>
> We note that our results suggest today's frontier language models are not sufficiently diverse even at the "ceiling" temperature of 1, and alternative, lower temperature settings (e.g., 0.3 or 0.7) would result in even less diversity among the generations.
>
> **3. Analysis of dataset diversity and distribution to avoid potential bias**
>
> We would be happy to provide additional analysis of dataset diversity and distribution. Could you elaborate on what specific kind of analysis you would like to see?

---

> > ### Comment · Reviewer_3g5a · 2025-06-09
> >
> > I acknowledged that I have carefully read the responses and my previous concerns have been solved. Regarding the bias, as all human reference answers come from eight volunteers who work or study at one institution. The paper notes that these writers “represent an approximate lower bound on actual human diversity,” yet their shared background can still imprint systematic preferences—for example, similar humor, reading habits, or car choices. Expanding the pool of annotators across age, discipline, and geography would give a stronger yardstick for comparing model diversity. I think this can be considered as a future work to maintain and expand this dataset. I improve my score according.

---

### Author Response · Authors · 2025-06-09

As the response period is ending soon, we would really appreciate if the reviewers can take a look at our responses and let us know if the concerns are addressed. Thank you.

\- Authors

---

### Decision · Program_Chairs · 2025-07-08

**Decision:**

Accept

**Comment:**

This paper presents NoveltyBench, to evaluate whether models can output distinct high quality outputs (i.e. test novelty or diversity) using prompting and testing for marginal utility of distinct answers. Authors evaluate 20 leading LLMs.

Reasons to Accept:

Multiple reviewers found the work important and fulfilling a real need in the field. They also note that the findings are surprising and interesting (e.g. larger models are often worse in terms of diversity/novelty).

Reasons to Reject:

Human reference answers come from a small number of volunteers. Reviewers also worried that evals are model-based which may introduce issues of accuracy and bias. Other suggestions include more analysis of post-training and diversity/distribution of dataset, as well as expansion of tasks would be useful.

The paper offers clear benefits to the field, but the work could be improved by further validating/improving model-based aspects of evaluation, and expanding on the existing analysis of model performance.